# Crosstalk between 17β-Estradiol and TGF-β Signaling Modulates Glioblastoma Progression

**DOI:** 10.3390/brainsci11050564

**Published:** 2021-04-28

**Authors:** Ana M. Hernández-Vega, Ignacio Camacho-Arroyo

**Affiliations:** Unidad de Investigación en Reproducción Humana, Instituto Nacional de Perinatología-Facultad de Química, Universidad Nacional Autónoma de México, México City, CP 11000, Mexico; anahdzvg@gmail.com

**Keywords:** 17β-estradiol (E2), transforming growth factor β (TGF-β), glioblastoma multiforme (GBM), epithelial–mesenchymal transition (EMT), estrogen receptor-α (ER-α), Smad2/3

## Abstract

Epithelial–mesenchymal transition (EMT) is an essential mechanism contributing to glioblastoma multiforme (GBM) progression, the most common and malignant brain tumor. EMT is induced by signaling pathways that crosstalk and regulate an intricate regulatory network of transcription factors. It has been shown that downstream components of 17β-estradiol (E2) and transforming growth factor β (TGF-β) signaling pathways crosstalk in estrogen-sensitive tumors. However, little is known about the interaction between the E2 and TGF-β signaling components in brain tumors. We have investigated the relationship between E2 and TGF-β signaling pathways and their effects on EMT induction in human GBM-derived cells. Here, we showed that E2 and TGF-β negatively regulated the expression of estrogen receptor α (ER-α) and Smad2/3. TGF-β induced Smad2 phosphorylation and its subsequent nuclear translocation, which E2 inhibited. Both TGF-β and E2 induced cellular processes related to EMT, such as morphological changes, actin filament reorganization, and mesenchymal markers (N-cadherin and vimentin) expression. Interestingly, we found that the co-treatment of E2 and TGF-β blocked EMT activation. Our results suggest that E2 and TGF-β signaling pathways interact through ER-α and Smad2/3 mediators in cells derived from human GBM and inhibit EMT activation induced by both factors alone.

## 1. Introduction

Malignant brain tumors represent a significant cause of cancer mortality and morbidity in the human population. These tumors contain cancer cells that deeply infiltrate through the surrounding brain tissue, making it challenging to complete surgical removal. Despite requiring a standard treatment of radiotherapy and chemotherapy, overall survival to five years is not larger than 35% [1]. Glioblastoma multiforme (GBM) is a grade IV astrocytoma that stands out as the most frequent malignant brain tumor with the worst prognosis [2,3,4].

Technological advances that allow determining the molecular profile have favored the genetic and epigenetic characterization of GBM, identifying biomarkers that can improve the precision of diagnosis and individualized treatment [5,6], including laser therapy [7,8]. The development and clinical validation of new compounds that specifically drive molecular targets and altered signaling pathways in GBM are under investigation. However, there are still many unknown aspects of the cells and molecular pathways that originate the various forms of GBM; their behavior and interactions within the tumor microenvironment; the plasticity mechanisms that give rise to different cellular phenotypes; among other pathological mechanisms that make GBM one of the most lethal malignant neoplasms.

The ability of tumor cells to infiltrate the nervous tissue and spread beyond the tumor’s visible edges are the main characteristic responsible for the high malignancy of GBM. Current treatments do not eliminate these invasive cells, leading to tumor recurrence and long-term therapy failure [9,10]. During the invasive process, GBM cells can reshape their environment by degrading the neural extracellular matrix (ECM) with the subsequent production of a new ECM containing mesenchymal components [11]. Structural signals from the ECM and soluble factors from surrounding cells regulate invasion mechanisms, including matrix remodeling, cytoskeletal reorganization, and cell phenotype transitions [12]. So far, a growing body of evidence supports the hypothesis that the epithelial–mesenchymal transition (EMT) plays a crucial role in the invasive phenotype of GBM [13,14,15,16,17].

EMT is a cellular mechanism in which cells lose their epithelial characteristics, including cell-cell adhesions and apicobasal polarity, and they simultaneously acquire the mesenchymal phenotype characterized by its migratory and invasive properties. The changes during EMT in cellular phenotype involve the remodeling of the intercellular junctions (adherents, tight, and desmosomes) and dynamic reorganization of the cytoskeleton [18,19,20]. EMT is activated by several signaling pathways that can cooperate in a cell-type-specific and context-dependent manner to activate the expression of EMT-inducing transcription factors (EMT-TFs), which coordinate epithelial gene repression and mesenchymal gene induction [21]. Some signaling pathways that activate the EMT-TFs in GBM include those mediated by transforming growth factor β (TGF-β) [22], receptor tyrosine-kinases (RTKs) [23,24,25], wingless-integration (Wnt)/beta-catenin [26,27], hypoxia [28,29], inflammatory cytokines such as interleukin 6 (IL-6) [30], and steroid hormones such as 17β-estradiol (E2) [31]. The complexity of the molecular network and the variables in each cell system provoke a partial or intermediate EMT program, resulting in very diverse cellular phenotypes [32]. Among the EMT-activating factors mentioned above, TGF-β stands out as the main inducer of EMT during embryonic development, fibrosis, and malignant progression [33].

TGF-β binds to type II serine/threonine kinase receptor (TβRII), leading to recruitment of TβRI to the complex and its subsequent phosphorylation, activating the canonical pathway through Sma-Mad related proteins (Smad) effectors, and non-canonical pathways driven by phosphatidylinositol-3-kinases (PI3K), mitogen-activated protein kinases (MAPK), and GTPases-RHO. The canonical signaling cascade is mediated by phosphorylation of receptor-regulated Smads (R-Smads), Smad2, and Smad3. The phosphorylated R-Smads form a complex with co-Smad, Smad4. Once formed, the complex translocates to the nucleus, where it recruits transcriptional coactivators such as p300 and cAMP response element-binding protein (CREB)-binding protein (CBP) to induce histones acetylation and activation of the expression of TGF-β target genes [34,35,36].

TGF-β signaling interacts with several pathways through the activation of many different effector proteins. Smad proteins have been shown to connect TGF-β to other signaling pathways, such as Wnt, Notch, Hippo, Hedgehog, MAPK, PI3K-Akt, and JAK-STAT pathways [37,38,39]. In estrogen-sensitive reproductive tissues, E2-induced signaling through the estrogen receptor α (ER-α) is related to the TGF-β signaling pathway [40,41,42,43]. Estrogens are steroid sex hormones that regulate various brain functions through their interaction with intracellular estrogen receptors (ERs), ER-α, and ER-β, widely distributed throughout the nervous tissue. Interestingly, high concentrations of E2 and a change in ERs expression levels have been detected in GBM biopsies, suggesting an essential role for these receptors in GBM pathology [44,45,46,47,48]. E2 through ER-α induces EMT in human GBM-derived cells [31]. EMT program activation by the E2 could vary in the microenvironment of GBM by interacting with the signaling pathways of other inducers of EMT, such as TGF-β.

As a first approach to understand TGF-β and E2 signaling interactions in GBM, in this study, we have evaluated the relationship between the canonical TGF-β pathway and E2-signaling through ER-α within the EMT context in human GBM-derived cells. Our results showed that: (1) TGF-β (10 ng/mL) and E2 (10 nM) negatively regulated the expression of the effectors Smad2, Smad3, and ER-α; (2) E2 inhibited the activation of canonical TGF-β pathway by decreasing the phosphorylation levels of Smad2 and the subsequent nuclear translocation of Smad2-Smad3 complex; (3) and finally, the inductor effects of both TGF-β and E2 on related EMT processes were mutually inhibited.

## 2. Materials and Methods

### 2.1. Cell Culture and Treatments

U251 and U87 cell lines derived from human GBM were acquired from American Type Culture Collection (ATCC, Manassas, VA, USA). Cells were cultured with Dulbecco’s modified Eagle’s medium (DMEM, Biowest, Nuaillé PDL, France) supplemented with 10% fetal bovine serum (FBS), 0.1 mM non-essential amino acids, 1 mM pyruvate, and 1 mM antibiotics (Biowest) in a humidified atmosphere with 5% CO_2_ at 37 °C. Before treatments, cells were seeded in DMEM no phenol red (Thermo Fisher Scientific, Waltham, MA, USA) supplemented with 10% hormone-free FBS (Thermo Fisher Scientific), 0.1 mM non-essential amino acids, 1 mM pyruvate, and 1 mM antibiotics (Biowest). Cells were treated with 10 nM E2 (Sigma-Aldrich, St. Louis, MO, USA), 10 ng/mL TGF-β (PeproTech, Rocky Hill, NJ, USA), and vehicle (0.01% cyclodextrin + 0.0001% BSA). 

### 2.2. Western Blotting

Cells were lysed with RIPA buffer supplemented with protease inhibitors (Sigma-Aldrich, St. Louis, MO, USA). The cell pellet was centrifuged at 20,817× *g* at 4 °C for 15 min, and the total protein was quantified with the Pierce Protein Assay reagent (Thermo Fisher Scientific, Waltham, MA, USA) in the NanoDrop 2000 spectrophotometer (Thermo Fisher Scientific). Total protein (30 µg) was boiled for 5 min with Laemmli sample buffer. Proteins were separated by SDS-polyacrylamide gel electrophoresis (PAGE) and transferred to polyvinylidene fluoride (PVDF) membranes (Merck, Kenilworth, NY, USA) under semi-dry conditions at room temperature, which were subsequently blocked with 5% bovine serum albumin (BSA) (Sigma-Aldrich) diluted in Tris-buffered saline-0.01%Tween (TBST). Membranes were incubated with primary antibodies: anti-ERα (2 µg/mL, rabbit monoclonal, ab3575, Abcam, Cambridge, UK), anti-Smad2/3 (1:500, rabbit monoclonal, 8685S), anti-phosphorylated Smad2 (1:500, rabbit monoclonal, 3108S) (Cell Signaling Technology, Danvers, MA, USA), anti-ZO-1 (0.6 µg/mL, rat monoclonal, R40.76: sc-33725), anti-N-cadherin (0.8 µg/mL, mouse monoclonal, D-4: sc-8424), anti-vimentin (0.4 µg/mL, mouse monoclonal, V9: SC-6260), and anti-α-tubulin (0.4 µg/mL, mouse monoclonal, A-6: sc-388103) (Santa Cruz Biotechnology, Dallas, TX, USA). After rinsing the membranes three times with TBST, they were incubated with secondary antibodies conjugated to horseradish peroxidase (HRP): anti-rabbit (0.06 µg/mL, goat polyclonal, 65–6120, Thermo Fisher Scientific), anti-mouse (0.013 µg/mL, purified recombinant mouse, sc:516102, Santa Cruz Biotechnology), and anti-rat (0.06 µg/mL, goat polyclonal, ab97057, Abcam) at room temperature for 45 min and rinsed with TBST three times. Chemiluminescence signals were detected exposing membranes to Kodak Biomax Light Film (Z370371, Sigma-Aldrich) with Super Signal West Femto Maximum Sensitivity Substrate reagent (34096, Thermo Fisher Scientific). Blots were captured using a digital camera (SD1400IS, Canon Inc., Ota, TY, Japan), and ImageJ software (National Institutes of Health, NIH, Bethesda, MD, USA) performed the densitometric analysis of blot images.

### 2.3. Immunofluorescence

Cells were fixed and permeabilizated with ice-cold 100% methanol at −20 °C for 10 min and then rinsed three times in cold phosphate-buffered saline (PBS). Cells were blocked in a PBS solution with 1% BSA for 60 min at 37 °C and incubated at 4 °C for 24 h with primary antibodies: anti-Smad2/3 (1:200, 8685S, rabbit monoclonal, Cell Signaling Technology, Danvers, MA, USA), and anti-actin (4 µg/mL, sc-1615, goat polyclonal, Santa Cruz Biotechnology, Dallas, TX, USA). Cells were rinsed three times in PBS and incubated with secondary antibodies: anti-rabbit (4 µg/mL, goat polyclonal Alexa Fluor 568, A11011, Thermo Fisher Scientific, Waltham, MA, USA), and anti-goat (8 µg/mL, donkey polyclonal FITC, Santa Cruz Biotechnology) at room temperature for 90 min. Cells were rinsed three times in PBS, and nuclei were stained for 7 min with 1 mg/mL Hoechst 33,342 (Thermo Fisher Scientific) and then rinsed three times in PBS. Cells were coverslipped with fluorescence mounting medium (Polysciences, Warrington, PA, USA) and visualized in a fluorescence microscope (Olympus Bx43, Shinjuku, TY, Japan). Images were captured at 400× magnification from five random fields in each independent experiment and analyzed in ImageJ software (National Institutes of Health, NIH, Bethesda, MD, USA). The percentage of nuclear colocalization was calculated considering the number of Hoechst-stained cells using the Cell Counter plugin in ImageJ software. The cell density in all experiments carried out was kept constant in both cell lines. In U251 cells, the cell number captured in each visual field of the microscope was approximately 200 ± 5%, while the average number of U87 cells captured was 45 ± 5% for the four experimental conditions.

### 2.4. Cell Morphology

Cell morphology was observed by phase-contrast microscopy (Olympus IX71, Shinjuku, TY, Japan) at 0, 48, and 72 h after adding each treatment and capturing six arbitrary fields with a 400× magnification for each independent experiment. Images were background and illumination corrected using Adobe Photoshop CS6 software (Adobe Systems Inc., San Jose, CA, USA). The morphological changes were determined using Image-Pro 10.0.6 software (Media Cybernetics Inc., Rockville, MD, USA) as previously done [31], quantifying the segmented cells’ geometric characteristics in the two-dimensional plane.

### 2.5. RT-qPCR

The TRIzol LS reagent (Thermo Fisher Scientific, Waltham, MA, USA) was used to extract the cells’ total RNA following the product’s standard protocol. The RNA concentration was determined with Nanodrop 2000 spectrophotometer, and its integrity was verified by electrophoresis with 1.5% agarose gel in Tris-Borate buffer using GreenSafe (NZYTech, Lisboa, PT, Portugal). Complementary DNA (cDNA) from 1 µg of RNA was obtained using Moloney Murine Leukemia Virus Reverse Transcriptase (M-MLV RT, 28025013, Thermo Fisher Scientific) following the provider’s standard protocol. The 18S ribosomal RNA (rRNA) was used as the housekeeping gene. Quantitative reverse transcription-polymerase chain reaction (RT-qPCR) experiments were performed with the FastStart DNA Master SYBR Green I kit (Roche, Basel, Switzerland) in a LightCycler 2.0 instrument (Roche) to gene amplification. The used primers were TJP1 (tight junction protein 1): FW-5′-gccattcccgaaggagttga-3′, RV-5′-atcacagtgtggtaagcg-3′; CDH2 (cadherin-2): FW-5′-ctggagacattggggacttc-3′, RV-5′-gagccactgccttcatagt-3′; and VIM (vimentin): FW-5′-ggaccagctaaccaacgaca-3′, RV-5′-aaggtcaagacgtgccagag-3′. rRNA18S [FW-5′-agtgaaactgcgaatggctc-3′, RV-5′-ctgaccgggttggttttgat-3′]. Relative expression was quantified by the comparative 2^∆∆Ct^ method [49].

### 2.6. Statistical Analysis

All data representing three independent experiments for each treatment were plotted and analyzed with GraphPad Prism 5.0 software (GraphPad, San Diego, CA, USA). Statistical analysis between comparable groups was performed using a one-way ANOVA with a Tukey post-hoc-test. Values of *p* < 0.05 were considered statistically significant.

## 3. Results

### 3.1. E2 and TGF-β Decreased the Expression of ER-α, Smad2, and Smad3 Proteins

We evaluated E2 and TGF-β effects on expressing some of their signaling components: ER-α, Smad2, and Smad3 in GBM cells (Figure 1). In U251 cells, we found that E2 decreased the expression of its ER-α and an effector of TGF-β, Smad3, while TGF-β only diminished ER-α expression. The treatment with E2 + TGF-β further decreased the expression of ER-α, Smad2, and Smad3. In U87 cells, E2 decreased ER-α, Smad2, and Smad3 expression, while TGF-β decreased ER-α and Smad3. Likewise, the conjunction of both inducers (E2 + TGF-β) further decreased the expression of the analyzed proteins. These results show that E2 and TGF-β regulate the expression of their signaling components.

### 3.2. E2 Decreased Smad2 Phosphorylation and Subsequent Nuclear Translocation of Smad2/3 Complex Induced by TGF-β

We evaluated the effect of E2 on TGF-β effector’s activation. Figure 2 shows that in U251 and U87 cells, TGF-β-induced phosphorylation of Smad2 at serine residues 465–467 was significantly inhibited by E2 treatment. We performed an immunofluorescence assay for evaluating Smad2/3 subcellular localization (Figure 3). Cells that received E2 showed the Smad2/3 complex was mainly located in the cytoplasm. In contrast, those treated with TGF-β mainly exhibited Smad2/3 complex in the nucleus; however, co-treatment of E2 + TGF-β decreased the nuclear localization of Smad2/3 in both U251 and U87 cells.

### 3.3. E2 and TGF-β Mutually Inhibited Their Effects on Morphological Changes Induced by Actin Filament Reorganization in GBM-Derived Cells

Some studies have shown that TGF-β [22] and E2 [31] promote EMT in human GBM-derived cell lines. We evaluated the effect of E2 and TGF-β co-treatment on cellular processes resulting from EMT activation on GBM cells, such as morphological changes and EMT molecular markers’ expression. Morphological characteristics of the captured cell images, such as circularity, XY box, aspect, and perimeter, were quantified to determine GBM cell morphology changes concerning the typical spindle shape of the mesenchymal phenotype (Figure 4). The quantized geometric parameters are defined by specific geometric measurements that the Image-Pro software (Media Cybernetics Inc., Rockville, MD, USA) performs around the contour or shape of the cells analyzed in the captured images. Values close to the unity of the circularity (ratio of figure area to the diameter of a circle around it) and box XY (ratio of the width to height of the bounding box of the figure outline.) measurements are characteristic of a polygonal shape. If these values decrease, it means that the morphology of the cell is going from being polygonal (epithelial) to being spindle shape (mesenchymal). Likewise, a high aspect (ratio between the major and minor axis of an ellipse surrounding the figure) and perimeter (length of the region surrounding the figure) denote a fusiform shape. At the beginning of the treatments, U251 and U87 cells were small and with polygonal morphology. E2 and TGF-β produced cytoplasmic projections, inducing a spindle-shaped appearance. However, the treatment of E2 + TGF-β inhibited the formation of these cytoplasmic projections prolongation by making them shorter. The quantitative analysis showed that the treatments with E2 and TGF-β induced a spindle shape in the cells since the circularity and the XY box decreased compared to the vehicle. Likewise, the aspect ratio and the perimeter length increased, which denotes the mesenchymal spindle shape. The co-treatment of E2 + TGF-β inhibited these changes in the cell geometric characteristics.

We performed an immunofluorescence assay to determine if cell morphological changes are directly related to actin filament reorganization. Figure 5 shows that U251 cells mainly presented actin in the cell cortex, with a dense reticulated mesh, giving the cells a polygonal shape with an epithelial phenotype. E2 and TGF-β induced long projections of actin filaments towards the leading edge, causing a mesenchymal morphology. However, the co-treatment of E2 + TGF-β inhibited the elongation of actin cytoskeleton projections.

### 3.4. E2 and TGF-β Mutually Inhibited Their Effects on EMT Markers Expression

The activation of EMT entails a series of changes in the cell molecular expression profile that can change the epithelial to the mesenchymal phenotype. We determined the epithelial marker ZO-1 and the mesenchymal markers N-cadherin and vimentin expression in U251 and U87 cells (Figure 6). In both cell lines, E2 increased the expression of the TJP1 gene and the protein it encodes, ZO-1. In contrast, TGF-β decreased this epithelial marker expression at the mRNA and protein levels only in U251 cells. The co-treatment of E2 and TGF-β nullified the effect of both inducers. E2 and TGF-β upregulated mRNA and protein expression of the mesenchymal markers, N-cadherin, and vimentin, but the co-treatment of E2 and TGF-β inhibited the upregulation of both markers.

These data showed that in human GBM-derived cells, the crosstalk between E2 and TGF-β interrupts both factors’ effects on cellular processes related to EMT activation, such as morphological changes, reorganization of actin filaments, and upregulation expression of molecular markers of the mesenchymal phenotype (Figure 7).

## 4. Discussion

This study determined the relationship between TGF-β and E2 signaling on EMT induction in human GBM-derived cells, finding that both mutually inhibit each other through the Smad2/3 and ER-α effectors. Dysregulation of these signaling pathways has been implicated in the development and progression of GBM.

The GBM tumor microenvironment is a complex system containing multiple cell types, including microglia, dendritic cells, neutrophils, lymphocytes, fibroblasts, astrocytes, oligodendrocytes, cancer stem cells (CSC), mesenchymal stem cells (MSC), and cancer cells [50,51,52]. These cells communicate through paracrine signaling mechanisms, mainly through the secretion of signals that induce nearby cells’ behavior changes. Therefore, the tumor microenvironment is immersed in an ECM enriched with various signaling factors that are decisive in the tumor progression of GBM [44,53,54,55]. The transduction of these factors forms networks resulting in the crosstalk among different signaling pathways [56,57,58,59]. Understanding the tumor microenvironment is currently recognized as a new challenge to improving brain tumors’ therapy [60].

Consequently, EMT induction in tumor microenvironmental conditions is a complex process since several stimuli can activate it, and signaling pathways of these stimuli are interconnected. Therefore, the outcome of EMT activation entirely depends on the context and cell type. Here, we focused on TGF-β and E2 found in high concentrations within the tumor [44,55]; both play an essential role in the progression of GBM [42,43,44,45,46,61,62,63], and signaling intersects to be inhibited in other cell types [41,42,43,64,65,66,67]. Furthermore, increasing evidence suggests that TGF-β signaling inhibition would provide new therapeutic options for GBM tumors in which TGF-β acts to promote proliferation, survival, and EMT [68,69,70,71].

In this work, we confirmed the known effects of TGF-β on EMT induction in U251 and U87 cells [22], and in turn, we evaluated these effects in conjunction with E2, which inhibited the effects of TGF-β on morphological changes and EMT markers expression in GBM cells. However, these results reveal some differences in TGF-β effects on regulating EMT markers expression between cell lines derived from GBM, probably due to differences in the expression profile between them [72,73]. We showed that similarly to other studies using breast cancer-derived cells, the blockage between both signaling pathways is because E2 decreases the Smad2 and Smad3 proteins expression [41,43]. However, it remains determined whether this decrease in expression directly regulates Smad2 and Smad3 gene expression or by mechanisms that induce these proteins’ degradation. Although so far, no study has described estrogen response elements in the promoters of Smad2 and Smad3. However, Ito et al. showed that ER-α leads to the termination of TGF-β signaling by forming a complex with Smad2/3 and the ubiquitin ligase Smurf, which increases the ubiquitination of Smad2/3 and its degradation by the proteasome [43]. As with Ito et al., we also show that E2 decreases phosphorylated Smad2 levels. It is thought that the decrease in Smad2 phosphorylation levels could be a consequence of the ER-α/Smad2/3 complex formation, which, when binding to ER-α, Smad2 undergoes a dephosphorylation mechanism [43]. Although in this work, we do not show the direct interaction between ER-α and Smad2/3, some studies have already demonstrated this complex’s formation [41,43,67]. Immunofluorescence assays against Smad2/3 shown that treating cells with E2 + TGF-β inhibits the translocation of Smad2/3 into the nucleus, indicating that decreased Smad2 phosphorylation disrupts the activity of R-Smads in the regulation of transcription of TGF-β target genes.

Besides, we also show that the co-treatment of E2 and TGF-β decreases ER-α expression. This inhibitory effect of E2 on ER-α expression was already demonstrated in our laboratory in other cell lines derived from human GBM [48]. E2 induces ER-α degradation through a self-regulatory mechanism known as ligand-dependent proteolysis, in which E2-induced receptor phosphorylation is a degradation signal through the ubiquitin-proteasome pathway [74,75,76]. This mechanism could explain the inversely proportional correlation between high concentrations of E2 and the decrease in ER-α as the degree of malignancy of astrocytomas increases [44]. Furthermore, TGF-β treatment has also decreased ER-α expression in breast cancer cells [77] and bronchial epithelial cells [64]. Our results indicate that treatment of E2 + TGF-β has a synergistic effect on ER-α expression inhibition.

Finally, we show that E2 and TGF-β induced EMT in glioblastoma cells. The independent treatments of E2 and TGF-β caused similar effects on the U251 and U87 cell mesenchymal morphology, which generally became more elongated in shape due to the reorganization of cortical actin to long dorsal tension fibers. However, these inducers’ effects on EMT markers expression were different since TGF-β decreased the epithelial marker expression (ZO-1) and increased the mesenchymal markers’ expression (N-cadherin and vimentin). In contrast, E2 increased the three markers’ expression. The latter highlights the ability of TGF-β to carry out a complete EMT in GBM cells, while E2 seems to activate a partial EMT. However, it is necessary to consider the determination of induced-EMT status by analyzing a more significant number of epithelial and mesenchymal markers. The co-treatment of both inducers (E2 + TGF-β) inhibited these effects. However, the consequences of this inhibition in cell proliferation, migration, and invasion are unknown. Much remains to be known about the dynamics between both signaling pathways within the tumor microenvironment. A more in-depth study of the molecular mechanisms of E2 signaling on GBM and its interaction with other signaling factors, such as TGF-β, in specific cellular contexts is necessary to understand the effects of several inducers in this tumor, which could provide new strategies in the treatment of GBM.

## Figures and Tables

**Figure 1 brainsci-11-00564-f001:**
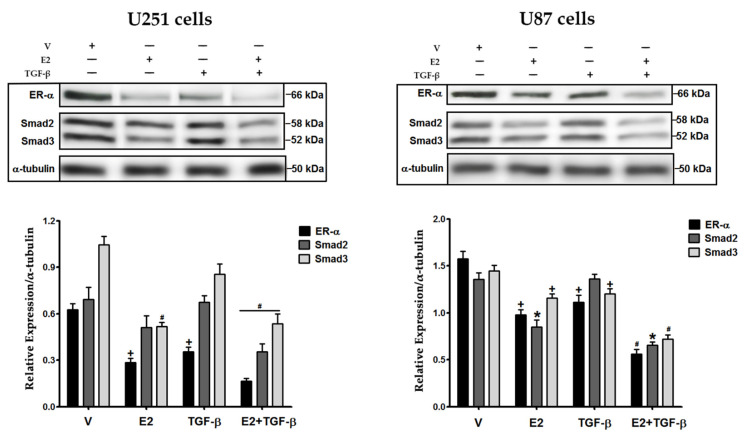
Effects of 17βestradiol (E2) and transforming growth factor β (TGF−β) on the expression of estrogen receptor α (ER-α), Smad2, and Smad3 in human GBM-derived cells. U251 and U87 cells were treated with vehicle (V, 0.01% cyclodextrin + 0.0001% BSA), E2 (10 nM), TGF-β (10 ng/mL) and E2 + TGF-β for 48 h. The expression of ER-α, Smad2, and Smad3 proteins was determined by Western Blot using α-tubulin as load control. Results are expressed as the mean ± standard error of the mean (SEM); *n* = 3; * *p* < 0.05 vs. all other groups; # *p* < 0.05 vs. V and TGF-β; + *p* < 0.05 vs. V.

**Figure 2 brainsci-11-00564-f002:**
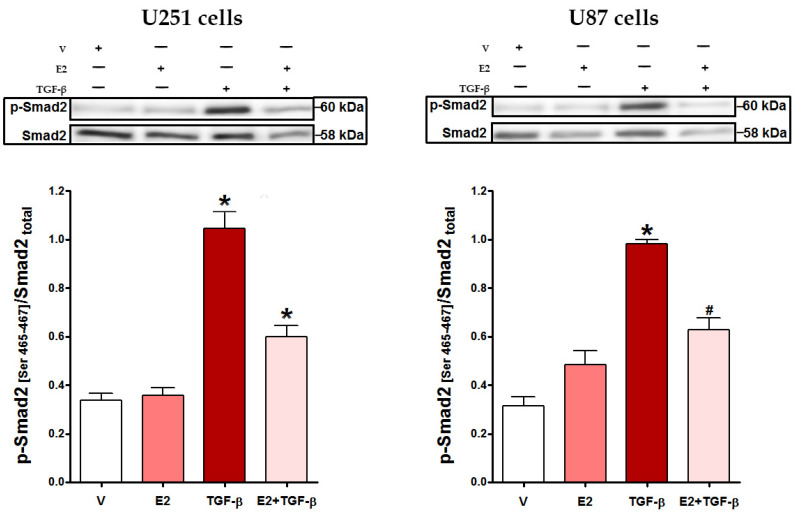
E2 effect on TGF-β-promoted Smad2 phosphorylation in human GBM-derived cells. U251 and U87 cells were treated with vehicle (V, 0.01% cyclodextrin + 0.0001% BSA), E2 (10 nM), TGF-β (10 ng/mL) and E2 + TGF-β for 48 h. Smad2 phosphorylation levels at serine residues 465–467 (*p*-Smad2) were determined by Western Blot. Densitometric analysis with their respective representative bands is shown. Results are expressed as the mean ± standard error of the mean (SEM); *n* = 3; * *p* < 0.05 vs. all other groups; # *p* < 0.05 vs. V.

**Figure 3 brainsci-11-00564-f003:**
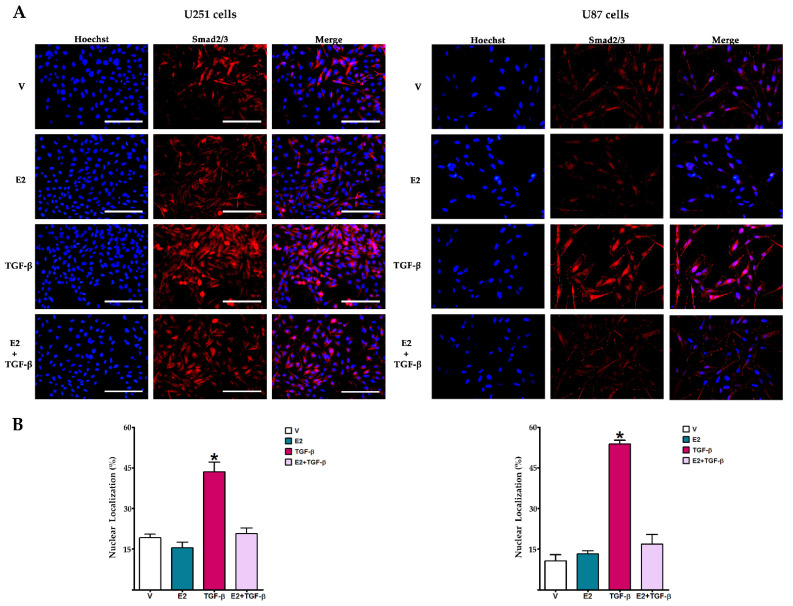
E2 effect on TGF-β-promoted nuclear translocation of Smad2/3 complex in human GBM-derived cells. (**A**) Smad2/3 immunostaining in U251 and U87 cells treated with vehicle (V, 0.01% cyclodextrin + 0.0001% BSA), E2 (10 nM), TGF-β (10 ng/mL) and E2 + TGF-β for 48 h. Scale white bars = 100 μm. (**B**) Nuclear colocalization of Smad2/3 complex. The percentage of nuclear colocalization was calculated considering the number of Hoechst-stained cells using the Cell Counter plugin in ImageJ software. Results are expressed as the mean ± standard error of the mean (SEM); *n* = 3; * *p* < 0.05 vs. all other groups.

**Figure 4 brainsci-11-00564-f004:**
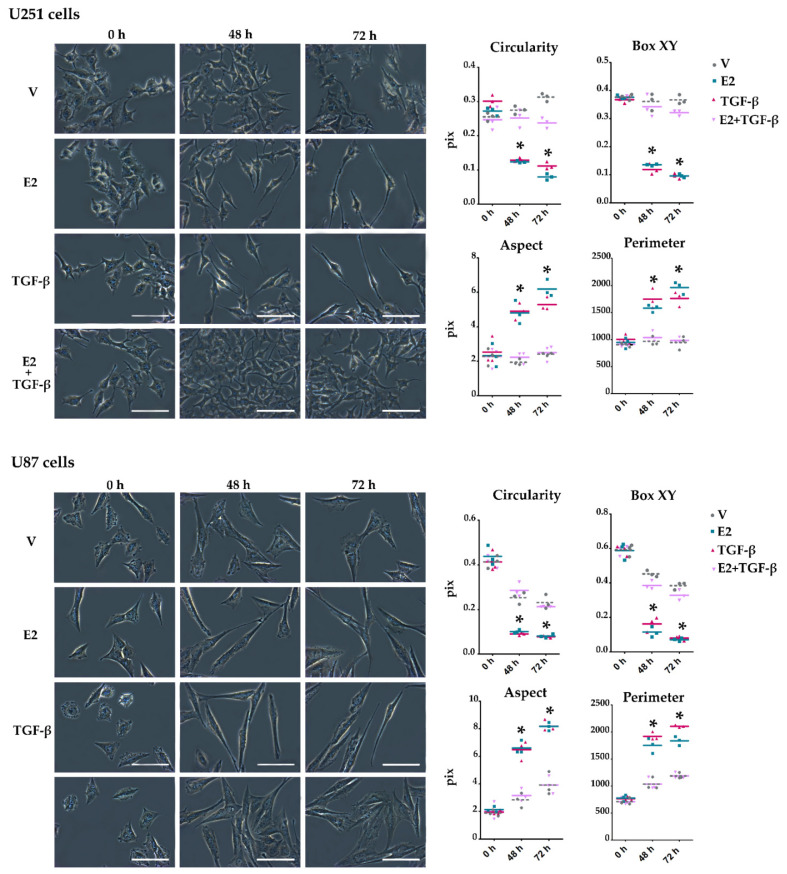
E2 and TGF-β-induced morphological changes in human GBM-derived cells. Representative images of U251 and U87 cells observed by phase-contrast microscopy at 0, 48, and 72 h after adding vehicle (V, 0.01% cyclodextrin + 0.0001% BSA), E2 (10 nM), TGF-β (10 ng/mL) and E2 + TGF-β. Scale white bars = 100 μm. Plots quantized geometric parameters in this study: circularity, box XY, aspect, and perimeter. Results are expressed as the mean ± standard error of the mean (SEM); *n* = 3; * *p* < 0.05 vs. V and E2 + TGF-β.

**Figure 5 brainsci-11-00564-f005:**
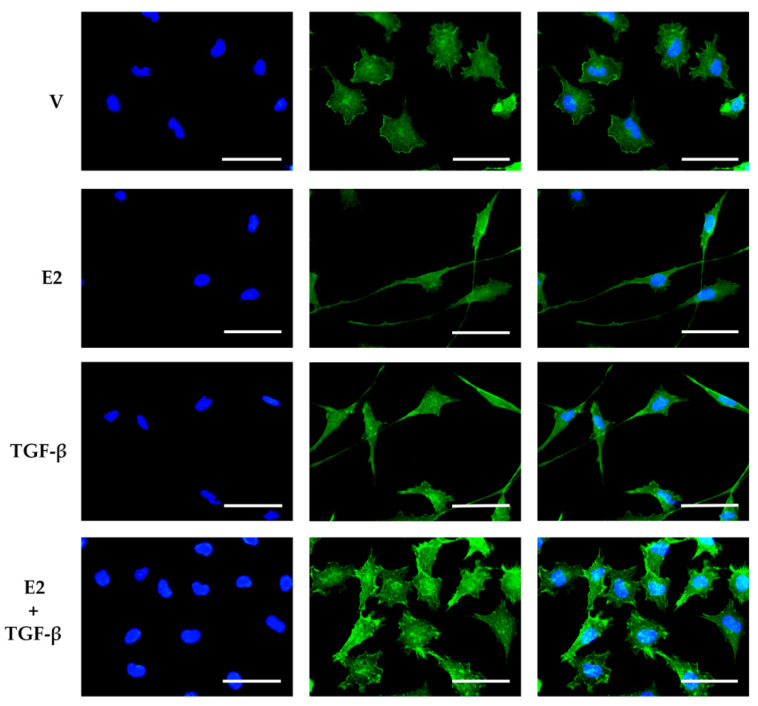
Effects of E2 and TGF-β on actin cytoskeleton reorganization in U251 cells. Actin immunostaining in U251 cells treated with vehicle (V, 0.01% cyclodextrin + 0.0001% BSA), E2 (10 nM), TGF-β (10 ng/mL) and E2 + TGF-β. Scale white bars = 30 μm. Representative images were captured under a fluorescence microscope at a magnification of 400×.

**Figure 6 brainsci-11-00564-f006:**
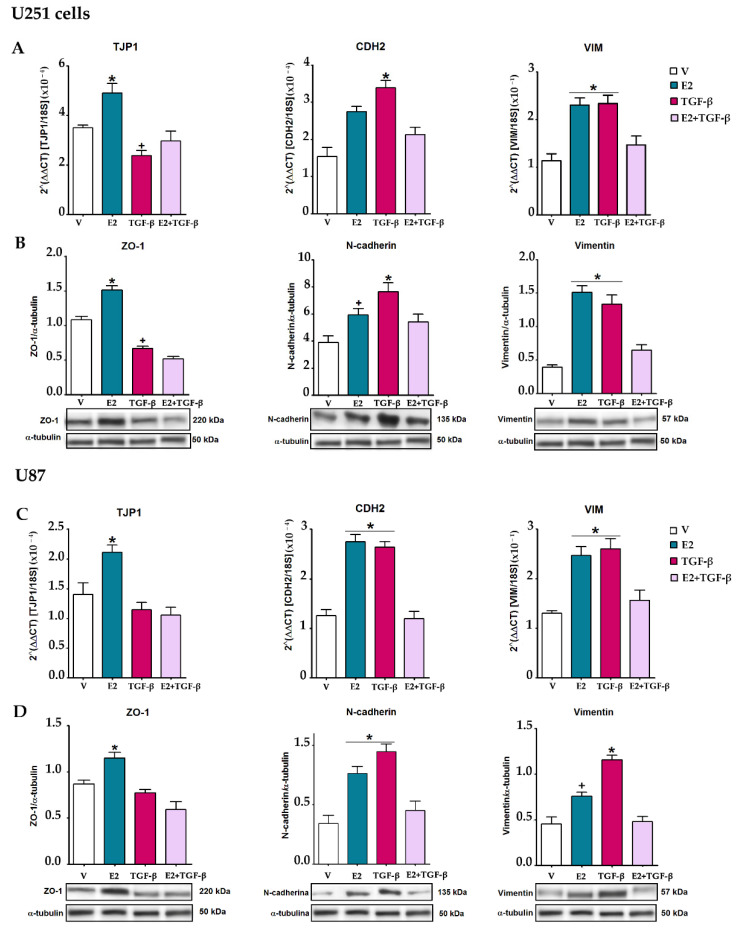
E2 and TGF-β-regulated epithelial–mesenchymal transition of human GBM-derived cells. U251 and U87 cells were treated with vehicle (V, 0.01% cyclodextrin + 0.0001% BSA), E2 (10 nM), TGF-β (10 ng/mL) and E2 + TGF-β for 48 h. (**A**,**C**) The expression of TJP1 (tight junction protein 1), CDH2 (cadherin-2), and VIM (vimentin) genes were quantified by RT-qPCR using the comparative method 2^ΔΔCt^ concerning the reference gene 18S rRNA. (**B**,**D**). The expression of zonula occludens 1 (ZO-1), N-cadherin, and vimentin proteins was determined by Western Blot. Densitometric analysis of EMT markers’ expression with their respective representative bands using α-tubulin as a load control is shown. Results are expressed as the mean ± standard error of the mean (SEM); *n* = 3; * *p* < 0.05 vs. V; + *p* < 0.05 vs. V and E2/TGF-β.

**Figure 7 brainsci-11-00564-f007:**
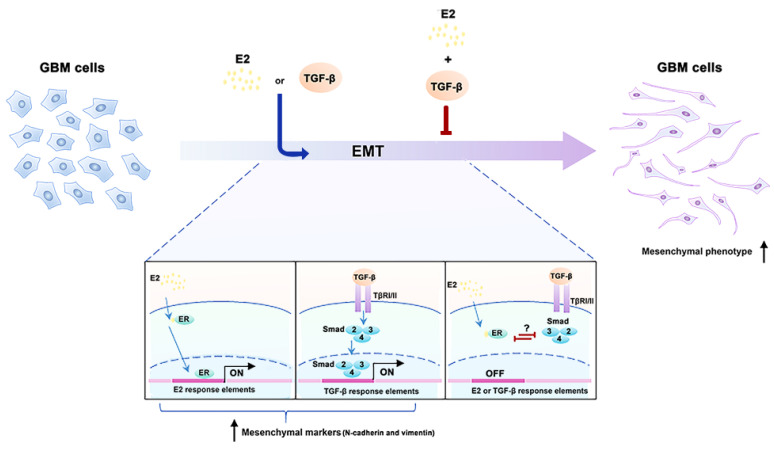
The molecular mechanism proposed for the activation or inhibition of EMT by E2 and TGF-β effects in human GBM cells. E2 and TGF-β promote EMT through ER and Smad, respectively. Crosstalk between ER and Smad inhibits EMT promoted independently by E2 and TGF-β. It is unknown whether the inhibition between E2 and TGF-β is through a direct or indirect interaction between ER and Smads. Epithelial–mesenchymal transition (EMT); 17β-estradiol (E2), estrogen receptor (ER); glioblastoma (GBM); transforming growth factor β (TGF-β); type I/II serine/threonine kinase receptor (TβRI/II).

## Data Availability

Not applicable.

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
