# Peer review of "Crosstalk between 17β-Estradiol and TGF-β Signaling Modulates Glioblastoma Progression"

_brainsci, 2021, doi:10.3390/brainsci11050564_

Round 1

Reviewer 1 Report

Well done. I think a xenograft evaluation and/or a trial E2 inhibitor would further validate the data. 

Author Response

Response to Reviewer 1 Comments

Well done. I think a xenograft evaluation and/or a trial E2 inhibitor would further validate the data.

Response:

We greatly appreciate your comments. We also believe that evaluating the crosstalk between E2 and TGF-β in a xenograft model and testing with E2 inhibitors would better validate the data shown in this work. We had planned to carry out such experiments; however, we could not carry them out due to the pandemic situation. Unfortunately, in Mexico, experimental 

Reviewer 2 Report

Authors have investigated the relationship between 17b-estradiol (E2) and TGF-b signaling pathways in U87 and U251 glioblastoma cells. Co-treatment of cells with E2 and TGF-b abrogated the TGF-b-mediated downstream signaling. E2 and TGF-b alone promoted epithelial-mesenchymal transition (EMT)-related morphological changes, whereas their combined treatment inhibited these changes. E2 and TGF-b individually had variable effects on EMT markers, which were negated by their co-treatment.

Major points:

  1. For the results showing effect of E2 or TGF-b alone or in combination on Smad2/3 immunofluorescence in Figure 3, authors have shown representative images that very clearly have significantly higher number of cells in the E2+ TGF-b condition, providing impression that increase in Smad2/3 nuclear localization in this condition is merely due to counting of more cells. How many cells per treatment condition were used for quantification? This is important information that’s missing from the methods/figure legends. Also, representative images with similar number of cells across conditions should be included.
  2. Authors observed partial EMT with E2 treatment in terms of its effect on TJP1 gene and encoded protein ZO-1. However, the morphological changes induced by E2 indicated complete EMT and were similar to TGF-b treatment, which induced complete EMT. How do authors explain the contrasting results of E2 on morphology and EMT markers?
  3. As mentioned in 2 above, analysis of additional epithelial markers such as E-cadherin (commonly used), cytokeratin, MUC-1 as well as a couple of mesenchymal markers such as α-SMA, Snail can provide better idea of the effect of E2 and TGF-b alone and in combination on EMT and may help understand the precise changes induced by these treatments.
  4. What are the effects of E2 and TGF-b on EMT-associated changes in the phenotypes of glioblastoma cells such as migration and invasion? Do combined treatment with E2+ TGF-b reverse the individual effects of E2 and TGF-b on these changes?

Minor points:

  1. “Content” of the components as mentioned in results of Figure 1 is odd language and should be changed to a scientific term such as “expression”.

Author Response

Response to Reviewer 2 Comments

Authors have investigated the relationship between 17b-estradiol (E2) and TGF-β signaling pathways in U87 and U251 glioblastoma cells. Co-treatment of cells with E2 and TGF-β abrogated the TGF-β-mediated downstream signaling. E2 and TGF-β alone promoted epithelial-mesenchymal transition (EMT)-related morphological changes, whereas their combined treatment inhibited these changes. E2 and TGF-b individually had variable effects on EMT markers, which were negated by their co-treatment.

Response: We highly appreciate your review. Your comments are very accurate and helped us to better analyze and discuss our results.

Major points:

Point 1: For the results showing effect of E2 or TGF-β alone or in combination on Smad2/3 immunofluorescence in Figure 3, authors have shown representative images that very clearly have significantly higher number of cells in the E2+TGF-β condition, providing impression that increase in Smad2/3 nuclear localization in this condition is merely due to counting of more cells. How many cells per treatment condition were used for quantification? This is important information that's missing from the methods/figure legends. Also, representative images with similar number of cells across conditions should be included.

Response 1: Thanks for the observation. The experimental conditions of the immunostaining were standardized in such a way that, at the time of starting the immunofluorescence assay, the number of cells was approximately the same in each treatment condition, since the cells receiving either E2 or TGF-β treatments, the number of cells tended to be higher compared to cells that only received vehicle or E2+TGF-β treatment. This observation in our trials is consistent with some studies showing that both E2 [1,2] and TGF-β [3-5] increase the cellular proliferation rate of GBM cells, which is also related to the EMT process.

The cell density in all experiments carried out was kept constant in both cell lines. In the case of U251 cells, the cell number captured in each visual field of the microscope was approximately 200±5%. Although cell number treated with E2 or TGF-β was higher than in-vehicle or E2+TGF-β, the cell number never exceeded 5% of the average among the four conditions in the experiments considered for this study. Likewise, the average number of U87 cells captured in each visual field was 45±5% for the four experimental conditions.

We greatly appreciate this observation as it allowed us to clarify these data. We replaced the representative images of the U87 cells in Figure 3, where the difference in the number of cells between the four treatments was observed. Besides, we have added more information in the Methods and Materials section (lines 160-163) and Figure 3 (lines 228-230) regarding immunofluorescence assay and the methodology performed to analyze the cellular localization of Smad.

Point 2: Authors observed partial EMT with E2 treatment in terms of its effect on TJP1 gene and encoded protein ZO-1. However, the morphological changes induced by E2 indicated complete EMT and were similar to TGF-b treatment, which induced complete EMT. How do authors explain the contrasting results of E2 on morphology and EMT markers?

Response 2: We appreciate the observations regarding this point as it allows us to improve the discussion of these results. The definition of partial EMT has become very important in recent years. In general, cells that activate EMT in adult tissues under pathological conditions commonly express both epithelial and mesenchymal markers and rarely show complete EMT. Cells with a hybrid epithelial-mesenchymal (E/M) phenotype have mixed epithelial and mesenchymal properties. The epithelial properties are determined by all those proteins that form intercellular junctions (adherent, tight, gap), while the mesenchymal properties are related to the proteins that promote cell migration and invasion, including the dynamic changes of the actin cytoskeleton. Experimental observations of the hybrid E/M phenotype have shown that the presence of intercellular junction complexes does not prevent the formation of typical mesenchymal spindle morphology that promotes migratory tumor processes. Therefore, the hybrid E/M phenotype allows tumor cells to move collectively with weak cell-cell junctions, allowing cancer cells to spread rapidly through the tissue [6-8].

Although it was observed that E2 increased the expression of both epithelial and mesenchymal markers, we cannot affirm the induction of a partial EMT since doing so requires the analysis of more epithelial and mesenchymal markers. Likewise, even though TGF-β is widely known to be one of the main inducers of complete EMT in certain stages of embryonic development and tissue regeneration, in tumor progression, it can induce partial EMT under certain specific-context conditions [9-11]. To better explain the latter's interpretation, we have modified the discussion that refers to this topic (lines: 369-370).

Point 3: As mentioned in 2 above, analysis of additional epithelial markers such as E-cadherin (commonly used), cytokeratin, MUC-1 as well as a couple of mesenchymal markers such as α-SMA, Snail can provide better idea of the effect of E2 and TGF-b alone and in combination on EMT and may help understand the precise changes induced by these treatments.

Response 3: The complex cellular changes that occur during EMT require the cooperation of many molecular factors. Given the complexity of the EMT program, the participation of EMT in specific molecular processes does not solely depend on a few molecular markers. Therefore, we fully agree that the analysis of other epithelial and mesenchymal markers would better let us understand the effects of E2 and TGF-β on GBM cells. Unfortunately, our plans to analyze the effects of E2 and TGF-β on the expression of other epithelial and mesenchymal markers were cut short in the middle of the last year by the current COVID-19 pandemic since it is not possible to enter the laboratories of our institution to conduct research that is not related to covid.

Regarding the analysis of other epithelial markers, although the E-cadherin marker represents the gold standard of epithelial phenotype, the use of this epithelial marker was dispensed due to its limited expression in GBM. E-cadherin expression has only been found in rare GBM species with pseudoepithelial differentiation, and its expression is low or null in various cell lines [12,13]. The same is true of most cytokeratins known as epithelial markers [14,15]. Recently, a multivariate analysis of the global expression profile determined that the most relevant epithelial markers to define an epithelial-like phenotype in GBM are TJP1 (ZO-1), MUC-1, and CLDN1 [16]. The studies mentioned above make it clear that epithelial markers for determining EMT are limited. If experiments were possible, MUC and CLDN1 would be used as additional epithelial markers in this study. Instead, many additional mesenchymal markers could be analyzed to validate our results in GBM cells. Nevertheless, as mentioned above, conducting experiments is impossible due to the current pandemic situation for COVID-19. However, it should be noted that even though only three EMT markers were analyzed, the results obtained in this study clearly show that the effects of E2 and TGF-β on the expression of these markers are mutually inhibited when both inducers are present. These observations open new perspectives regarding more detailed studies that explain the changes induced at the level of expression of EMT markers due to the crosstalk between E2 and TGF-β.

Point 4: What are the effects of E2 and TGF-b on EMT-associated changes in the phenotypes of glioblastoma cells such as migration and invasion? Do combined treatment with E2+ TGF-b reverse the individual effects of E2 and TGF-b on these changes?

Response 4: We recently showed that E2 promotes migration and invasion processes in GBM  cells [17]. Likewise, TGF-β has also been found to increase GBM migration and invasion [18]. However, the combined effect between E2 and TGF-β on GBM cell migration and invasion is unknown. Initially, our objective was to test the effect between E2 and -β on the migration and invasion of GBM-derived cells. However, these experiments could not be performed for the close of our institution's laboratories due to the current COVID-19 pandemic. Therefore, experiments showing the effects of E2 and TGF-β on GBM cell migration and invasion remain to be determined, as mentioned in our discussion (lines: 371-377).

Minor points:

Point 1: "Content" of the components as mentioned in results of Figure 1 is odd language and should be changed to a scientific term such as "expression".

Response 1: We greatly appreciate this observation. We have changed the word "content" to "expression" in the results of Figure 1 (lines: 196-202, 205, 207, and 289).

References

  1. González-Arenas, A.; Hansberg-Pastor, V.; Hernández-Hernández, O.T.; González-García, T.K.; Henderson-Villalpando, J.; Lemus-Hernández, D.; Cruz-Barrios, A.; Rivas-Suárez, M.; Camacho-Arroyo, I. Estradiol increases cell growth in human astrocytoma cell lines through ERα activation and its interaction with SRC-1 and SRC-3 coactivators. Biochim Biophys Acta 2012, 1823, 379-386, doi:10.1016/j.bbamcr.2011.11.004.
  2. Castracani, C.C.; Longhitano, L.; Distefano, A.; Anfuso, D.; Kalampoka, S.; La Spina, E.; Astuto, M.; Avola, R.; Caruso, M.; Nicolosi, D., et al. Role of 17β-Estradiol on Cell Proliferation and Mitochondrial Fitness in Glioblastoma Cells. J Oncol 2020, 2020, 2314693, doi:10.1155/2020/2314693.
  3. Bruna, A.; Darken, R.S.; Rojo, F.; Ocaña, A.; Peñuelas, S.; Arias, A.; Paris, R.; Tortosa, A.; Mora, J.; Baselga, J., et al. High TGFbeta-Smad activity confers poor prognosis in glioma patients and promotes cell proliferation depending on the methylation of the PDGF-B gene. Cancer Cell 2007, 11, 147-160, doi:10.1016/j.ccr.2006.11.023.
  4. Roberts, A.B.; Wakefield, L.M. The two faces of transforming growth factor beta in carcinogenesis. Proc Natl Acad Sci U S A 2003, 100, 8621-8623, doi:10.1073/pnas.1633291100.
  5. Platten, M.; Wick, W.; Weller, M. Malignant glioma biology: role for TGF-beta in growth, motility, angiogenesis, and immune escape. Microsc Res Tech 2001, 52, 401-410, doi:10.1002/1097-0029(20010215)52:4<401::Aid-jemt1025>3.0.Co;2-c.
  6. Yang, J.; Antin, P.; Berx, G.; Blanpain, C.; Brabletz, T.; Bronner, M.; Campbell, K.; Cano, A.; Casanova, J.; Christofori, G., et al. Guidelines and definitions for research on epithelial-mesenchymal transition. Nat Rev Mol Cell Biol 2020, 21, 341-352, doi:10.1038/s41580-020-0237-9.
  7. Jolly, M.K.; Boareto, M.; Huang, B.; Jia, D.; Lu, M.; Ben-Jacob, E.; Onuchic, J.N.; Levine, H. Implications of the Hybrid Epithelial/Mesenchymal Phenotype in Metastasis. Front Oncol 2015, 5, 155, doi:10.3389/fonc.2015.00155.
  8. Saitoh, M. Involvement of partial EMT in cancer progression. J Biochem 2018, 164, 257-264, doi:10.1093/jb/mvy047.
  9. Xu, J.; Lamouille, S.; Derynck, R. TGF-beta-induced epithelial to mesenchymal transition. Cell Res 2009, 19, 156-172, doi:10.1038/cr.2009.5.
  10. Tian, X.J.; Zhang, H.; Xing, J. Coupled reversible and irreversible bistable switches underlying TGFβ-induced epithelial to mesenchymal transition. Biophys J 2013, 105, 1079-1089, doi:10.1016/j.bpj.2013.07.011.
  11. Wendt, M.K.; Allington, T.M.; Schiemann, W.P. Mechanisms of the epithelial-mesenchymal transition by TGF-beta. Future Oncol 2009, 5, 1145-1168, doi:10.2217/fon.09.90.
  12. Perego, C.; Vanoni, C.; Massari, S.; Raimondi, A.; Pola, S.; Cattaneo, M.G.; Francolini, M.; Vicentini, L.M.; Pietrini, G. Invasive behaviour of glioblastoma cell lines is associated with altered organisation of the cadherin-catenin adhesion system. J Cell Sci 2002, 115, 3331-3340.
  13. Noh, M.G.; Oh, S.J.; Ahn, E.J.; Kim, Y.J.; Jung, T.Y.; Jung, S.; Kim, K.K.; Lee, J.H.; Lee, K.H.; Moon, K.S. Prognostic significance of E-cadherin and N-cadherin expression in Gliomas. BMC Cancer 2017, 17, 583, doi:10.1186/s12885-017-3591-z.
  14. Oh, D.; Prayson, R.A. Evaluation of epithelial and keratin markers in glioblastoma multiforme: an immunohistochemical study. Arch Pathol Lab Med 1999, 123, 917-920, doi:10.1043/0003-9985(1999)123<0917:Eoeakm>2.0.Co;2.
  15. Goswami, C.; Chatterjee, U.; Sen, S.; Chatterjee, S.; Sarkar, S. Expression of cytokeratins in gliomas. Indian J Pathol Microbiol 2007, 50, 478-481.
  16. Takashima, Y.; Kawaguchi, A.; Yamanaka, R. Promising Prognosis Marker Candidates on the Status of Epithelial-Mesenchymal Transition and Glioma Stem Cells in Glioblastoma. Cells 2019, 8, doi:10.3390/cells8111312.
  17. Hernández-Vega, A.M.; Del Moral-Morales, A.; Zamora-Sánchez, C.J.; Piña-Medina, A.G.; González-Arenas, A.; Camacho-Arroyo, I. Estradiol Induces Epithelial to Mesenchymal Transition of Human Glioblastoma Cells. Cells 2020, 9, doi:10.3390/cells9091930.
  18. Joseph, J.V.; Conroy, S.; Tomar, T.; Eggens-Meijer, E.; Bhat, K.; Copray, S.; Walenkamp, A.M.; Boddeke, E.; Balasubramanyian, V.; Wagemakers, M., et al. TGF-β is an inducer of ZEB1-dependent mesenchymal transdifferentiation in glioblastoma that is associated with tumor invasion. Cell Death Dis 2014, 5, e1443, doi:10.1038/cddis.2014.395.

Reviewer 3 Report

This paper reported a very interesting topic about the interaction between the E2 and TGF-β signaling components in brain tumors. Please look at these point:

  1. Lines 115-142: "Western Blotting. Cells were lysed with radioimmunoprecipitation assay..." this paragraph is hard to read, can the authors try to shorten it?
  2. Lines 238-241: "The quantitative analysis showed that the treatments with E2 and TGF-β induced a spindle shape in the cells since the circularity and the XY box (width/height) decreased compared to the vehicle. Likewise, the aspect ratio (major/minor axis) and the perimeter length increased, which denotes the mesenchymal spindle shape" Can the authors try to explain these notions better ?
  3. Lines 39-43: "However, there are still many unknown aspects of the cells ... mechanisms that make GBM one of the most lethal malignant neoplasms" In the introduction section, it should be add also the role of radiotherapy or laser therapy in GBM. Please look at these two recent ref.:  TGF-β in radiotherapy: Mechanisms of tumor resistance and normal tissues injury. Pharmacol Res. 2020 May;155:104745. doi: 10.1016/j.phrs.2020.104745.    -    Survival outcomes in patients with recurrent glioblastoma treated with Laser Interstitial Thermal Therapy (LITT): A systematic review. Clin Neurol Neurosurg. 2020 Aug;195:105942. doi: 10.1016/j.clineuro.2020.105942.
  4. Lines 321-323: "However, it remains determined whether this decrease in expression directly regulates Smad2 and Smad3 gene expression or by mechanisms that induce these proteins' degradation" Does this paper have any limitations? if yes, please add.
  5. Lines 296-300: "About GBM microenvironment". The glioblastoma microenvironment is really the first milestone for new groundbreaking therapeutic strategies. Please discuss here more Ref: TGF-β Signaling and Its Targeting for Glioma Treatment; E-Century Publishing Corporation: Madison, WI, USA, 2015; Volume 5, pp. 945–955. 

Author Response

Response to Reviewer 3 Comments

This paper reported a very interesting topic about the interaction between the E2 and TGF-β signaling components in brain tumors. Please look at these points:

Point 1: Lines 115-142: "Western Blotting. Cells were lysed with radioimmunoprecipitation assay..." this paragraph is hard to read, can the authors try to shorten it?

Response 1: Thanks for the recommendation. We have modified the text (line: 117).

Point 2: Lines 238-241: "The quantitative analysis showed that the treatments with E2 and TGF-β induced a spindle shape in the cells since the circularity and the XY box (width/height) decreased compared to the vehicle. Likewise, the aspect ratio (major/minor axis) and the perimeter length increased, which denotes the mesenchymal spindle shape" Can the authors try to explain these notions better?

Response 2: Thanks for the observation. We have added a more precise explanation about this morphological analysis and its results (lines: 240-248, 252-255).

Point 3: Lines 39-43: "However, there are still many unknown aspects of the cells ... mechanisms that make GBM one of the most lethal malignant neoplasms" In the introduction section, it should be add also the role of radiotherapy or laser therapy in GBM. Please look at these two recent ref.:  TGF-β in radiotherapy: Mechanisms of tumor resistance and normal tissues injury. Pharmacol Res. 2020 May;155:104745. doi: 10.1016/j.phrs.2020.104745.    -    Survival outcomes in patients with recurrent glioblastoma treated with Laser Interstitial Thermal Therapy (LITT): A systematic review. Clin Neurol Neurosurg. 2020 Aug;195:105942. doi: 10.1016/j.clineuro.2020.105942.

Response 3: Thanks for the recommendation. We have added the importance of laser therapy in GBM with the recommended references (lines: 37-38, 401-406).

Point 4: Lines 321-323: "However, it remains determined whether this decrease in expression directly regulates Smad2 and Smad3 gene expression or by mechanisms that induce these proteins' degradation" Does this paper have any limitations? if yes, please add.

Response 4: Initially, one of our goals was to determine whether E2-inhibited TGF-β signaling was due to direct binding with Smad2/3 and ER-α. However, these experiments could not be carried out due to the closure of our institution's laboratories due to the current COVID-19 pandemic. Despite this, we showed that E2 decreases phosphorylation levels of Smad2 and nuclear translocation of the Smad2/ Smad3 complex, which indicates that E2 is mainly interfering with the canonical signaling of TGF-β which is the central theme of our study. Therefore, because of the COVID pandemic, we could not perform several experiments, which is the main limitation of our work.

Point 5: Lines 296-300: "About GBM microenvironment". The glioblastoma microenvironment is really the first milestone for new groundbreaking therapeutic strategies. Please discuss here more Ref: TGF-β Signaling and Its Targeting for Glioma Treatment; E-Century Publishing Corporation: Madison, WI, USA, 2015; Volume 5, pp. 945–955.

Response 5: Thanks for the recommendation. We also believe that understanding the glioblastoma tumor microenvironment is the key to developing new and more effective therapeutic strategies, so we have added this topic to the discussion, including the recommended reference (lines: 317-318, 524-525).

Round 2

Reviewer 3 Report

Authors solved all my criticisms.